# Impact of Stain Normalization on Pathologist Assessment of Prostate Cancer: A Comparative Study

**DOI:** 10.3390/cancers15051503

**Published:** 2023-02-27

**Authors:** Massimo Salvi, Alessandro Caputo, Davide Balmativola, Manuela Scotto, Orazio Pennisi, Nicola Michielli, Alessandro Mogetta, Filippo Molinari, Filippo Fraggetta

**Affiliations:** 1Biolab, PoliTo^BIO^Med Lab, Department of Electronics and Telecommunications, Politecnico di Torino, Corso Duca degli Abruzzi 24, 10129 Turin, Italy; 2Department of Medicine and Surgery, University Hospital of Salerno, 84084 Fisciano, Italy; 3Pathology Unit, Humanitas Gradenigo Hospital, Corso Regina Margherita 8, 10153 Turin, Italy; 4Technology Transfer and Industrial Liaison Department, Politecnico di Torino, Corso Duca degli Abruzzi 24, 10129 Turin, Italy; 5UOC di Anatomia Patologica, ASP Catania P.O. “Gravina”, 95041 Caltagirone, Italy

**Keywords:** digital pathology, prostate cancer, stain normalization, color quality, Gleason score

## Abstract

**Simple Summary:**

Prostate cancer is the second most diagnosed cancer in men worldwide, with an estimated 1,276,000 new cases and 359,000 deaths in 2018. It is graded using the Gleason system into five grade groups of increasing tumor aggressiveness. However, diagnosis is hampered by a relatively high rate of inter- and intra-observer variability. Currently, the reduction of the perceived color variability is performed by physical quality controls, such as subjective assessment by visual inspection and comparison between laboratories. However, slides from different laboratories and even from different batches of the same laboratory may show significant color variations. The stain normalization procedure helps to standardize the stain color appearance of a digital image with respect to a reference image. In this study, we investigated the impact of the stain normalization process on prostate cancer biopsies from the pathologist’s perspective.

**Abstract:**

In clinical routine, the quality of whole-slide images plays a key role in the pathologist’s diagnosis, and suboptimal staining may be a limiting factor. The stain normalization process helps to solve this problem through the standardization of color appearance of a source image with respect to a target image with optimal chromatic features. The analysis is focused on the evaluation of the following parameters assessed by two experts on original and normalized slides: (i) perceived color quality, (ii) diagnosis for the patient, (iii) diagnostic confidence and (iv) time required for diagnosis. Results show a statistically significant increase in color quality in the normalized images for both experts (*p* < 0.0001). Regarding prostate cancer assessment, the average times for diagnosis are significantly lower for normalized images than original ones (first expert: 69.9 s vs. 77.9 s with *p* < 0.0001; second expert: 37.4 s vs. 52.7 s with *p* < 0.0001), and at the same time, a statistically significant increase in diagnostic confidence is proven. The improvement of poor-quality images and greater clarity of diagnostically important details in normalized slides demonstrate the potential of stain normalization in the routine practice of prostate cancer assessment.

## 1. Introduction

Prostate cancer (PCa) is the second most commonly diagnosed cancer in men worldwide, with an estimated 1,276,000 new cases and 359,000 deaths in 2018 [1]. The most recent world health organization (WHO) classification of urinary and male genital tumors [2] defines a five-tiered classification scheme [3] to better correlate the pre-existing Gleason score [4] with accumulating data on prognosis. The five resulting Grade Groups (GG), as shown in Figure 1, range from 1 (corresponding to Gleason score 3 + 3 = 6) to 5 (corresponding to Gleason scores 4 + 5, 5 + 4, and 5 + 5) and are related to different clinical outcomes and treatment choices [5,6,7,8,9,10]. The usage of virtual microscopy and whole-slide images (WSIs) to diagnose prostate cancer has been shown by multiple groups to be adequate and non-inferior to conventional microscopic diagnosis [11,12,13]. However, PCa diagnosis is hampered by a relatively high rate of inter-observer and intra-observer discordance, by optical microscopy and virtual microscopy alike [14,15,16].

Digital pathology is a rapidly growing field of pathology based on the adoption of new technologies such as WSIs, digital workflow and computer-aided diagnosis (CAD) [17]. Artificial intelligence (AI)-based tools are being used to assist pathologists and clinicians at large in the diagnosis, prognostication and treatment of patients [18,19]. The importance for a standardization of both procedures and reagents in clinical practice is emphasized in the study of Lyon [20]. However, current technology does not allow for complete standardization due to stain fading over time and the variability of the manual sectioning process. Currently, the reduction of perceived color variability and its impact on diagnosis is performed by procedural and physical quality controls, such as subjective quality assessment by visual inspection and comparison between laboratories. However, slides from different laboratories and even from staining batches of the same laboratory may be subjected to significant color variations [21,22]. These staining variations in tissue appearance make the quantitative analysis of histological tissue a complex process [23].

The quality of the histological slide plays a key role in the pathologist’s diagnosis [24,25]. A staining that is too weak or too intense can hide important diagnostic details, reducing confidence, increasing the time for analysis, and possibly causing misdiagnosis. In recent years, the stain normalization procedure has been proposed to address this issue in digital slides. The stain normalization procedure helps to standardize the stain color appearance of a digital image (also denoted as source image) with respect to a reference image (also denoted as target image). The target image is a single image chosen by the pathologist with the most optimal visual appearance and tissue staining. Stain normalization changes the chromatic information of the source image so that the color appearance matches the target image. In this way, the stain variability of histological images can be reduced. Several algorithms for color normalization have been proposed [21,26], and different studies have shown that normalized images enable improved performances of various algorithms for digital pathology image analysis [27,28,29,30]. However, to the best of our knowledge, no study has investigated the possible impact of the color normalization process on the PCa diagnostic routine on biopsies or the qualitative improvement due to this process from the pathologist’s perspective.

In this study, we assess the impact of the normalization process within the diagnostic routine and specifically during the assessment of PCa. The analysis is focused on the evaluation of several parameters, such as the perceived color quality, the diagnosis for the patient, the confidence and the time for diagnosis, for both original and normalized WSIs. This paper is organized as follows: in Section 2, the description of materials and methods is provided, in Section 3, the experimental results are reported, and they are discussed in Section 4. In Section 5, the conclusions are summarized.

## 2. Materials and Methods

### 2.1. Dataset

In this study, WSIs of prostate tissue, derived from formalin-fixed and paraffin-embedded samples stained with hematoxylin and eosin (H&E) were selected from our previous works [29,31]. The selection was carried out with the aim of simulating the high stain variability of histopathological slides, ranging from lighter to darker stain color images with some minors suboptimal staining such as reddish, greyish color and weak staining. A total of 93 WSIs scanned from the same department of oncology and from different patients were collected. Bad-quality glass slides and images for which pathologists requested a second opinion with immunohistochemical markers were removed from the analysis. WSIs were digitized with three digital slide scanners: 15 slides were acquired with Aperio AT2 at 200× magnification (0.467 µm/pixel), 36 slides were scanned with Aperio GT450 at 400× magnification (0.233 µm/pixel), and 42 slides were captured with Hamamatsu NanoZoomer S210 at 200× magnification (0.467 µm/pixel).

### 2.2. Clinical Study Description

Stain normalization was performed for each WSI in our dataset, resulting in 2 subsets: (i) the original dataset (i.e., the 93 scanned WSIs) and (ii) the normalized dataset (i.e., the corresponding 93 normalized WSIs). We performed the normalization process with STAINS—STAndardIzation & Normalization of histological Slides—tool (AEQUIP S.r.l., Turin, Italy), an improved version of our previously published algorithm [26]. STAINS tool is based on a fully automated approach of stain normalization of H&E histopathological WSIs and consists of the following steps: (i) tissue detection to distinguish relevant information from background regions; (ii) illuminant correction based on white balancing; (iii) separation of hematoxylin and eosin color channels and (iv) normalization of both channels to adapt chromatic profile of target image. This process preserves local structures and enhances the contrast between histological tissue and background. The output of STAINS, regardless of the input image format, is a 200× tiled pyramidal TIFF image with JPEG lossy compression (quality factor = 80). For most tissue types and diagnostic tasks, optical magnification of 200× is considered sufficient to identify relevant biological features. The JPEG compression was used to save memory since the size of a single uncompressed WSI is usually over 2.5 GB [32]. The normalization of a single WSI using the STAINS tool took on average 4 min. An example of a prostate WSI, processed with the stain normalization tool, is shown in Figure 2.

Two different pathologists denoted as Pathologist 1 (P1) and Pathologist 2 (P2) from two medical centers and with different years of experience in pathology (P1 with 12 years and P2 with 6 years of experience) were involved in this study. Both pathologists analyzed original and normalized images using QuPath open-source software [33]. Firstly, P1 observed and evaluated the original dataset, and P2 observed and evaluated the normalized one. After 3 months, the process was repeated in the reverse direction, i.e., P1 was asked to evaluate the normalized images while P2 was asked to assess the original ones. The identification name of each WSI was randomly generated to prevent each pathologist from rechecking the case and the diagnosis previously performed. Each pathologist, for each slide, provided the following evaluations:Assessment of perceived stain color quality: it is quantified with a numerical scale from 1 to 10, where 1 indicates a low-quality image and 10 a high-quality image.Diagnosis of the given slide: if tumor is present, it is graded according to the Gleason Grade Groups [3,4] from 1 to 5. If no tumor is present on the given slide, a score of 0 is assigned.Confidence in the given diagnosis: rated subjectively from 1 to 10, where 1 indicates a low degree of reliability in diagnosis and 10 denotes a high degree of confidence. Operatively, a high-confidence diagnosis occurs when the pathologist thinks that the given slide is sufficient to perform a diagnosis, whereas low confidence indicates that the pathologist is not fully convinced by the appearance of the examined slide and would resort to recuts or immunohistochemical analysis.Time required for diagnosis: it is expressed in seconds and indicates the time taken by the pathologist to examine the image in order to decide the diagnostic classification. It was measured from the time when the image is opened (i.e., when the pathologist starts examining the image) to the time when the diagnosis is formulated; after that, the pathologist stops examining the image and writes down the diagnosis. The time required for image loading, thus, was not considered.

The workflow followed in this study is summarized in Figure 3.

The results of this study are divided into two main sections: color quality analysis and PCa evaluation. In the former, the pathologists’ evaluation of color quality is analyzed, while in the latter, the evaluations of diagnosis, confidence and time for diagnosis are reported. In this way, we aim to assess the impact of the normalization process on the pathologist’s perceived quality and diagnosis process. For the assessment of stain color quality, confidence and time for diagnosis, the values of the original and normalized WSIs are compared for each pathologist using a paired *t*-test with a 5% significance level. In addition, we assessed the agreement between pathologists in the evaluation of diagnosis using the quadratic weighted Cohen’s kappa coefficient.

## 3. Results

### 3.1. Evaluation of the Color Quality

In this section, we compare color quality data of the 93 original WSIs and the corresponding normalized WSIs. Both pathologists, denoted as P1 and P2 in the following, evaluated the original and normalized dataset. We analyzed the distribution of color quality given by each pathologist. Figure 4 shows color quality values with boxplots for overall distributions. The median values for normalized image distributions (8/10 for P1 and 7/10 for P2) are greater than the median values of the original image distributions (6/10 for P1 and 5/10 for P2). In addition, for both overall distributions, 25th and 75th percentiles related to normalized images are higher than corresponding percentiles of original images. The increase in color quality in the normalized images is statistically significant (paired-sample *t* test, *p* < 0.0001 for both P1 and P2). This result shows that the perceived stain color quality is higher in normalized images for both pathologists.

Examples of the normalization process performed on original images of suboptimal quality (score < 4/10) are shown in Figure 5. In particular, the color normalization process improves the quality of images showing reddish (Figure 5—sample #1) or grayish (Figure 5—sample #2) staining or too weak staining with poor contrast between cellular structures (Figure 5—sample #3). In addition to showing a better staining, normalized images have a more repeatable and standardized color appearance.

### 3.2. Assessment of Prostate Cancer: Diagnosis, Time and Confidence

This section describes the results related to PCa evaluation: pathologists were asked to provide the diagnosis expressed as Grade Group (GG), confidence and time for diagnosis on the original 93 WSIs and the corresponding normalized ones.

Firstly, we analyzed the agreement between pathologists’ diagnoses for both original and normalized images. The two experts’ intra- and inter-rater agreements of GG values were investigated through the quadratic weighted Cohen’s kappa coefficient. The intra-rater agreement (Cohen’s kappa) for P1 (original vs. normalized) is 0.8908 and for P2 (original vs. normalized) is 0.8959. The inter-rater agreement (P1 vs. P2) is 0.8507 for original images and 0.7946 for normalized ones. Furthermore, Figure 6 shows the confusion matrices on GG scores to evaluate the consensus of both pathologists on original and normalized images.

The clinical process is also evaluated in terms of time employed to formulate a diagnosis. In Figure 7, the overall distributions of time for diagnosis for original WSIs and normalized ones are represented for both pathologists. Specifically, for P1, the median values are 72.0 s (original) vs. 65.0 s (normalized), and for P2, the values are 39.6 s (original) vs. 30.6 s (normalized). In addition, for P1, 33% of original images have a time for diagnosis higher than 90.0 s, while only 11% of normalized images have a time for diagnosis higher than 90.0 s. For P2, 14% of original images have a time for diagnosis higher than 90.0 s, while only 2% of normalized images have a time for diagnosis higher than 90.0 s. The decrease in diagnosis time in the normalized images is statistically significant (paired-sample *t* test, *p* = 0.00015 for P1 and *p* < 0.0001 for P2).

Finally, we compared the diagnostic confidence related to original and normalized WSIs. We evaluated the overall distributions of the confidence data in Figure 8. Specifically, for P1, median value, 25th and 75th percentiles related to normalized images are higher than the ones related to original images, while for P2, median value and 75th percentile of normalized images are equal to the ones of the original image, while the 25th percentile is higher for normalized images. In addition, the minimum values of confidence related to original images (i.e., 3/10 for P1 and 1/10 for P2) are lower than minimum values related to normalized ones (i.e., 7/10 for P1 and 6/10 for P2). The increase in confidence in the normalized images is statistically significant (paired-sample *t* test, *p* < 0.0001 for P1 and *p* = 0.00028 for P2).

A summary of data related to the assessment of PCa is presented. According to the first pathologist (P1), mean and standard deviation values of time for diagnosis are 77.9 ± 26.6 and 69.9 ± 27.6 s for original and normalized images, respectively, while for the second expert (P2) are 52.7 ± 32.2 and 37.4 ± 24.4 s, respectively. Regarding confidence score, the original images have shown 7.0 ± 1.8 and 6.2 ± 2.4 for P1 and P2, respectively, while the corresponding normalized ones reported a confidence value of 8.2 ± 1.2 and 7.2 ± 1.2, respectively. No relevant influence about two different input magnifications (i.e., 200× and 400×) was observed on the evaluation parameters: color quality, time for diagnosis and confidence in diagnosis.

## 4. Discussion

Staining variability is one of the problems affecting the quality of WSIs and can negatively influence the diagnostic process. These problems can be solved by using a stain-normalization procedure. This technique can change the color information of a digital image so that its color appearance matches the color appearance of the reference image chosen by the pathologist [21]. In our previous study, we proved the superiority of normalized images with respect to the corresponding original ones in terms of perceived image quality and absence of clinically significant artifacts in a multi-center and multi-tissue context [25].

In this study, we analyzed the impact of the normalization process on the clinical routine related to the diagnosis of PCa. For this purpose, we analyzed data collected from two pathologists: one pathologist (denoted as P1) performed the evaluation on 93 original scanned WSIs, while the second expert (denoted as P2) assessed Pca on the corresponding normalized WSIs. After 3 months, the process was repeated in the reverse direction, i.e., P1 evaluated the normalized images, and P2 analyzed the original ones. The specialists were asked to evaluate the images in terms of stain color quality, diagnosis, confidence, and time for diagnosis. To the best of our knowledge, no study analyzed the normalization process from a clinical perspective.

The experimental results are divided into two sections: perceived color quality and PCa evaluation. Regarding the color quality analysis, Figure 4 shows that the quality of normalized images is on average better than the quality of the corresponding original ones for both pathologists (6/10 vs. 8/10 for P1 and 5/10 vs. 7/10 for P2). The normalization process has a positive impact on the quality perceived by pathologists, and it is useful when the quality of starting original images is not satisfactory. Color normalization is also evaluated in terms of its impact on the PCa diagnostic process by analyzing tumor classification, confidence, and time for diagnosis. When comparing the diagnoses formulated by each pathologist on the original vs. normalized images, most are completely concordant or only slightly discordant (difference of a single class). This is shown in Figure 6 and is demonstrated by the weighted Cohen’s kappa test (0.8908 and 0.8959 for P1 and P2, respectively). The inter-rater agreement is also very high (weighted Cohen’s kappa: 0.8507 for original images and 0.7946 for normalized ones). The diagnosis process is also evaluated in terms of time. Results show that pathologists take longer to classify the tumor on original images with respect to the normalized ones (77.9 s vs. 69.9 s for P1 and 52.7 s vs. 37.4 s for P2). From these data, we can state that the time for diagnosis is directly related to the quality of the tissue slide, and the normalization process, by improving color quality, enables faster diagnosis. Finally, pathologists were asked to assess their confidence on diagnosis. For both pathologists, mean values of confidence increase for normalized slides with respect to original ones (i.e., 8.2/10 vs. 7.0/10 for P1 and 7.2/10 vs. 6.2/10 for P2). We can therefore assume that confidence in diagnosis is subjected to less variability due to the stain normalization process and that confidence is strongly related to stain quality. Paired analysis (original vs. normalized, for each pathologist) of the time required for diagnosis and the confidence of diagnosis similarly reveal interesting data. The time required for diagnosis is significantly shorter on normalized images compared to original images, for both pathologists (*p* < 0.001). This can be explained by the greater clarity of diagnostically important details in normalized images. Examination of low-quality original images proved difficult for both pathologists, who reported difficulties in assessing details such as presence and size of nucleoli, chromatin quality and cytoplasm shape and color. In some cases, the architecture proved too difficult to examine, leading on one side to overestimation of benign mimickers of cancer, and on the other side to underestimation of minor foci of small/malformed Gleason pattern 4 glands, which were missed in the original images but were correctly identified in normalized ones, as shown in Figure 6. For the same reasons, the confidence in diagnosis was found to be significantly higher in normalized images compared to original images, for both pathologists (*p* < 0.001).

This study has different limitations related to the number of WSIs analyzed (n = 93) and pathologists involved (n = 2), which used a bioimage software (QuPath [33]) for research purposes, not for clinical use. Another limitation consists in the dataset composition, which originated from the same medical center and was selected to cover the high variability of color staining for the evaluation of the normalization process, instead of reporting consecutive cases belonging to same batches to represent typical clinical practice. In addition, the choice of target image for the normalization process can be investigated in a future work. We plan to employ further efforts to collect a larger number of WSIs and involve more experts with different years of experience in digital pathology with the possibility to select a specific target image for each of them. The choice of target image may depend on the pathologist’s opinion according to the clinical expertise and with the objective of improving the diagnosis. We also intend to extend the study to more tissues/pathologies (e.g., breast cancer; colon cancer; etc.) and expand the use of the normalization process on different hematoxylin (e.g., Gill 1,2,3; Harris; Mayer; etc.) and eosin (e.g., watery vs. alcoholic) solutions and other histological staining techniques, e.g., periodic acid Schiff (PAS) and trichrome staining.

## 5. Conclusions

In this study, we analyzed the impact of the stain normalization process to the diagnosis of prostate cancer from a clinical perspective. In summary, we have shown how a digital color normalization procedure can support the physician during diagnosis formulation, reducing analysis time and increasing the confidence of diagnosis.

## Figures and Tables

**Figure 1 cancers-15-01503-f001:**
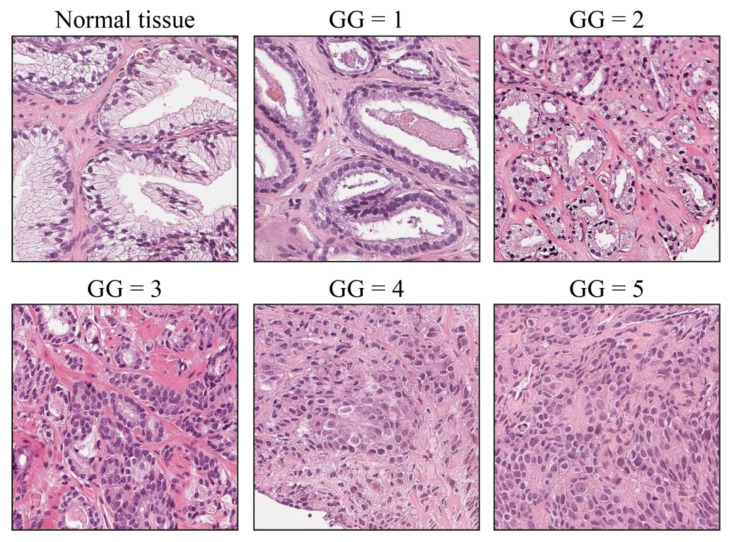
Prostate cancer grade groups. Normal prostatic tissue is shown for reference. Grade Group 1 (GG1) is a low-grade tumor with well-formed individual glands. With increasing GG, glands become fused, disorganized, solid, cribriform, or they are not formed at all (GG4, GG5).

**Figure 2 cancers-15-01503-f002:**
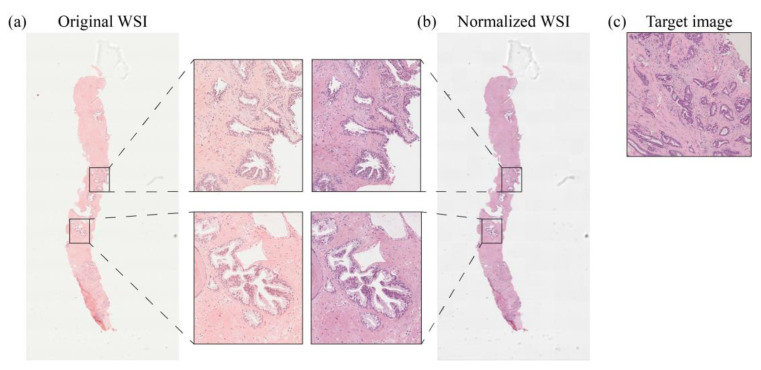
Stain normalization process for a prostate WSI. Original slide (**a**) and normalized slide (**b**) with zoomed-in views showing the effects at high magnification of color normalization with respect to target image (**c**).

**Figure 3 cancers-15-01503-f003:**
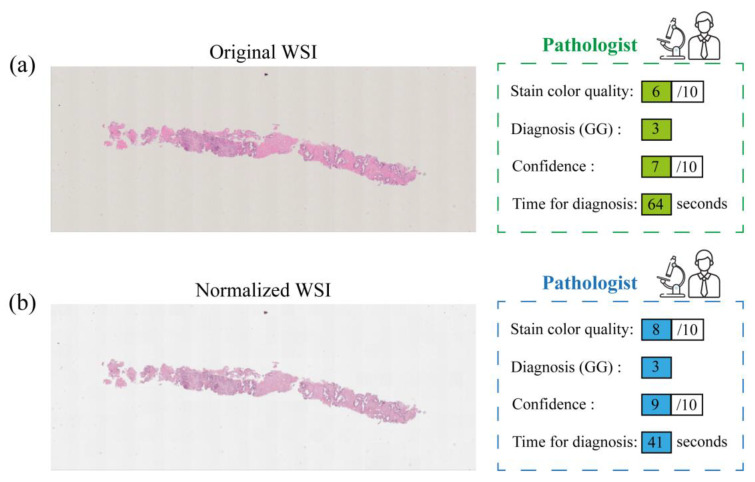
Workflow adopted in this study: an example of original WSI (**a**) and the corresponding normalized slide (**b**) processed by STAINS tool. Both pathologists (P1 and P2) provided evaluations on stain color quality, diagnosis, confidence, and time for diagnosis, for both datasets, i.e., the original images (an example of pathologist’s evaluation is reported in green) and the corresponding normalized ones (an example of pathologist’s evaluation is reported in blue).

**Figure 4 cancers-15-01503-f004:**
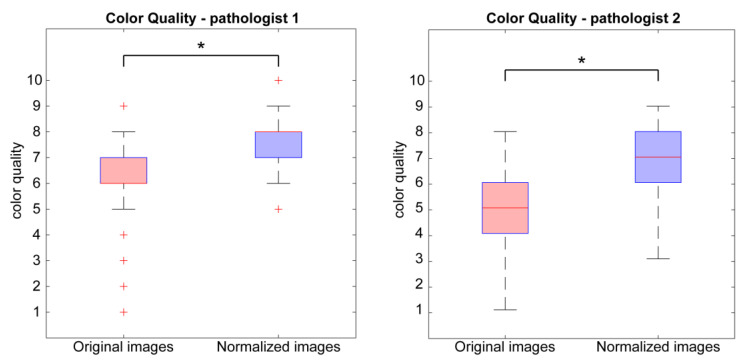
Boxplots of distributions of color quality values of original images (red) and normalized images (blue). The evaluation of the first pathologist (P1) is shown on the left, and the evaluation of the second pathologist (P2) is represented on the right. The plus symbol (+) indicates outliers in the distribution, while the asterisk (*) denotes a statistically significant difference between original and normalized images.

**Figure 5 cancers-15-01503-f005:**
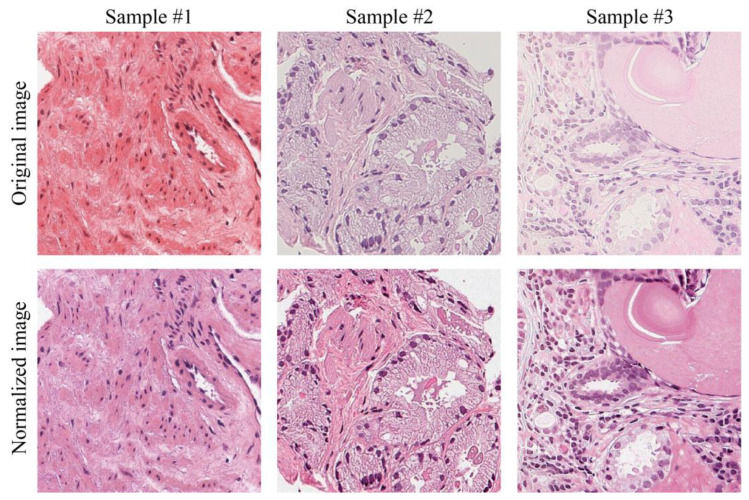
Visual performance of normalization process. In the first row, there are examples of tiles with suboptimal staining: reddish (sample #1), greyish color (sample #2) and weak staining (sample #3). The second row shows the corresponding normalized tiles.

**Figure 6 cancers-15-01503-f006:**
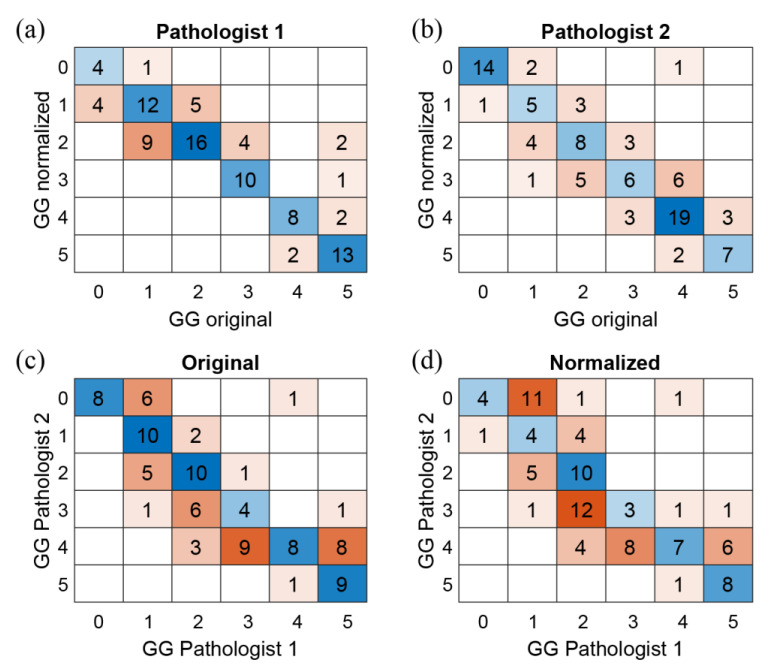
Confusion matrices on Gleason Grade Group (GG). First row shows the intra-agreement of P1 (**a**) and P2 (**b**) on the GG values of the original and normalized images. Second row shows the inter-agreement (P1 vs. P2) on the original (**c**) and normalized (**d**) WSIs.

**Figure 7 cancers-15-01503-f007:**
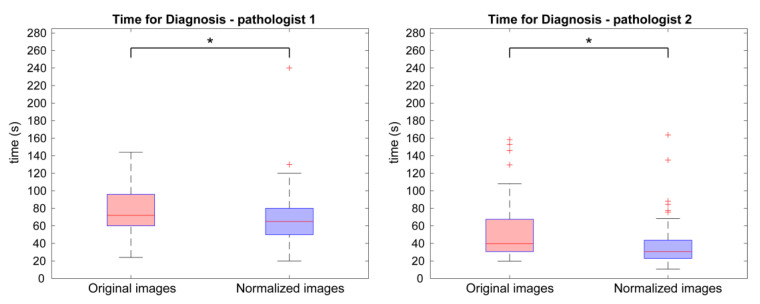
Comparison between time for diagnosis of the original and normalized WSIs. Boxplot of diagnosis time values of original (red) and normalized images (blue) according to the evaluations of the first pathologist (P1) on the left and second one (P2) on the right. The plus symbol (+) indicates outliers in the distribution, while the asterisk (*) denotes a statistically significant difference between original and normalized images.

**Figure 8 cancers-15-01503-f008:**
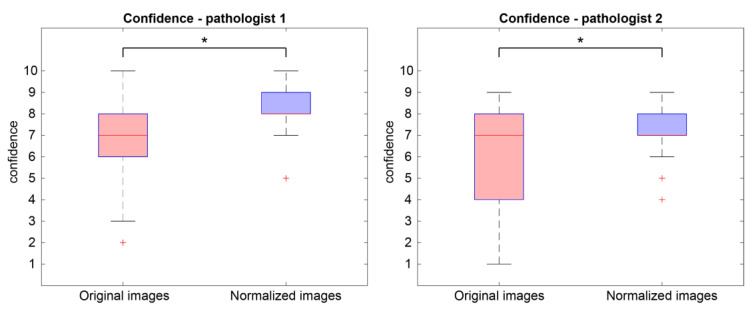
Boxplots of confidence values. Overall distribution of confidence values of original (red) and normalized images (blue) according to the evaluations of the first pathologist (P1) on the left and second one (P2) on the right. The plus symbol (+) indicates outliers in the distribution, while the asterisk (*) denotes a statistically significant difference between original and normalized images.

## Data Availability

The datasets used and/or analyzed during the current study are available from the corresponding author on reasonable request.

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
