# Peer review of "Impact of Stain Normalization on Pathologist Assessment of Prostate Cancer: A Comparative Study"

_cancers, 2023, doi:10.3390/cancers15051503_

Round 1
Reviewer 1 Report
Page 3:
- How 93 WSIs were selected?
If they were selected specially for this study, - probably, including uncommon, bad stains, then they may not represent typical practice and the results (e.g., time/case) achieved might be exaggerated. Authors should exactly describe the selection process and explain how the selection bias was avoided and in how much the slides represent a typical practice. If they do not represent typical practice (e.g., consecutive cases), then authors should state this as limitation.
- Do all slides stem from one department? If yes, the authors should state this as a limitation.
- How the target image for normalization was selected?
- What is the principle underlying STAINS normalization?
- What is a format of output for STAINS – the same for original file (e.g., iSyntax) or it would be saved in open-source format (BigTiff or OME.TIFF)? If so, how the authors deal with the original color scheme integrated into the original image?
- Is any compression/re-compression applied to the resulting image? If yes, how authors address the influence of this factor on any downstream results?
- Any conventional normalization algorithm (Macenko, Reinhardt, Vahadane) will produce (sometimes heavy) color artifacts: fat tissue, regions with low tissue content, blood etc. Was this an issue in this study? Could authors provide representative examples and estimate the number of affected cases?
- Authors should state as limitation that QuPath is a tool not validated/certified for diagnostic purposes.
- Authors should provide the details of which displays were used by pathologists – clinical grade or not, size, whether they were color calibrated or not.
Reviewer 2 Report
I would like to have an information about the influence of magnification (μm/pixel) on all described parameters (57 WSIs scanned at 200x vs 36 WSIs at 400x).
Do you have slides without stain normalization, where pathologist was quicker and more confident than on normalized slides? If yes, please explain why.
